

# Estimating salt content of vegetated soil at different depths with Sentinel-2 data

Yinwen Chen[1,*], Yuanlin Qiu[2,*], Zhitao Zhang[2,3], Junrui Zhang[2,3], Ce Chen[2,3], Jia Han[2,3] and Dan Liu[2]

[1] Department of Foreign Languages, Northwest A&F University, Yangling, Shaanxi, China
[2] College of Water Resources and Architectural Engineering, Northwest A&F University, Yangling, Shaanxi, China
[3] Key Laboratory of Agricultural Soil and Water Engineering in Arid and Semiarid Areas, Ministry of Education, Northwest A&F University, Yangling, Shaanxi, China
[*] These authors contributed equally to this work.

Corresponding author
Zhitao Zhang, zhitaozhang@126.com

## ABSTRACT

The accurate and timely monitoring of the soil salt content (SSC) at different depths is the prerequisite for the solution to salinization in the arid and semiarid areas. Sentinel-2 has demonstrated significant superiority in SSC inversion for its higher temporal, spatial and spectral resolution, but previous research on SSC inversion with Sentinel-2 mainly focused on the unvegetated surface soil. Based on Sentinel-2 data, this study aimed to build four machine learning models at five depths (0~20 cm, 20~40 cm, 40~60 cm, 0~40 cm, and 0~60 cm) in the vegetated area, and evaluate the sensitivity of Sentinel-2 to SSC at different depths and the inversion capability of the models. Firstly, 117 soil samples were collected from Jiefangzha Irrigation Area (JIA) in Hetao Irrigation District (HID), Inner Mongolia, China during August, 2019. Then a set of independent variables (IVs, including 12 bands and 32 spectral indices) were obtained based on the Sentinel-2 data (released by the European Space Agency), and the full subset selection was used to select the optimal combination of IVs at five depths. Finally, four machine learning algorithms, back propagation neural network (BPNN), support vector machine (SVM), extreme learning machine (ELM) and random forest (RF), were used to build inversion models at each depth. The model performance was assessed using adjusted coefficient of determination ($R^2_{adj}$), root mean square error (RMSE) and mean absolute error (MAE). The results indicated that 20~40 cm was the optimal depth for SSC inversion. All the models at this depth demonstrated a good fitting ($R^2_{adj} \approx 0.6$) and a good control of the inversion errors (RMSE < 0.16%, MAE < 0.12%). At the depths of 40~60 cm and 0~20 cm the inversion performance showed a slight and a great decrease respectively. The sensitivity of Sentinel-2 to SSC at different depths was as follows: 20~40 cm > 40~60 cm > 0~40 cm > 0~60 cm > 0~20 cm. All four machine learning models demonstrated good inversion performance ($R^2_{adj} > 0.46$). RF was the best model with high fitting and inversion accuracy. Its $R^2_{adj}$ at five depths were between 0.5 to 0.68. The SSC inversion capabilities of all the four models were as follows: RF model > ELM model > SVM model > BPNN model. This study can provide a reference for soil salinization monitoring in large vegetated area.

## INTRODUCTION

Soil salinization has been an important factor leading to crop yield reduction and land degradation in arid and semiarid areas (*Harti et al., 2016*). Efficient and accurate monitoring of soil salt content (SSC) on a large scale is the key to tackle this problem. Among the varied monitoring methods, satellite remote sensing has become increasingly prevailing.

So far, different satellites and methods have been applied to soil salinization monitoring. *Lobell et al. (2010)* first used MODIS data for regional-scale soil salinity assessment and reduced the effect of temporally dynamic factors using the mean of the enhanced vegetation index (EVI) for a 7-year period. With consideration of the effects of precipitation, crop type, and soil texture, *Scudiero, Skaggs & Corwin (2014)* assessed the SSC based on the average of multi-year Landsat 7 data, and obtained reliable results. *Wu et al. (2014b)* mapped soil salinity mainly with Landsat ETM+ and MODIS multi-year data, and achieved reliable salinity prediction results in vegetated and non-vegetated areas, respectively. IKONOS data were used to analyze the Pearson correlation coefficient between broadband indices and soil salinity, the results indicated that the correlation depended on the environmental conditions (soil, vegetation cover and density), and vegetation indices performed better in densely vegetated areas (*Allbed, Kumar & Aldakheel, 2014*). Landsat 8 data were used to construct 12 VI-SI (vegetation indices-salinity indices) feature spaces based on the information of bare soil and vegetation. Results showed that MSAVI-SI$_1$(modified soil adjust vegetation index-salinity index) can greatly improve the dynamic and periodical monitoring of soil salinity (*Guo et al., 2019*). These studies on the relationship between multiple satellite data and soil salinity have provided a good basis for regional SSC assessment. However, each of the above satellites has demonstrated such defects as low spatial resolution or small spectral range. Sentinel-2 has shown certain advantages because it simultaneously has high temporal and spatial resolution, which enable more detailed and higher-frequency monitoring for practical applications. Additionally, Sentinel-2 can obtain the red-edge region of vegetation spectrum, which can provide more effective data for vegetation growth monitoring.

Scholars have conducted some research on SSC inversion with Sentinel-2 data. *Wang et al. (2020)* estimated soil salinity using the machine learning model, Cubist. By comparing the two SSC distribution maps (at the depth of 0∼20 cm), they found that Sentinel-2 outperformed Landsat 8 in accuracy. *Davis, Wang & Dow (2019)* and *Gorji et al. (2020)* also discovered that Sentinel-2 had great potential for SSC inversion. *Taghadosi, Hasanlou & Eftekhari (2019)* established two models (multiple linear regression and support vector regression) using Sentinel-2 images, which had good performance in SSC inversion in the unvegetated areas. *Wang et al. (2019)* created multiple spectral indices based on Sentinel-2 data and developed an RF-PLSR model to estimate SSC. The above studies on SSC inversion with Sentinel-2 data were mostly concentrated in the surface soil. *Ramos et al. (2020)* evaluated soil salinity at the depth of 0∼1.5m via multiple stepwise regression based on multi-year Sentinel-2 data, and obtained relatively high prediction accuracy (the coefficient of determination ranged from 0.63 to 0.91). This study lays a groundwork for

soil salinity estimation at root depth based on Sentinel-2 data. However, the evaluation of SSC at different root depths in the vegetated soil remains to be investigated.

In addition, the machine learning algorithms have been widely used for SSC inversion and water resources management (*Wang et al., 2019*). *Chen et al. (2015)* studied the accuracy of multiple linear regression (MLR), back propagation neural network (BPNN), and support vector machine (SVM) in soil salinity estimation using hyperspectral data, and found that BPNN and SVM were more accurate than MLR. When studying the hybrid particle swarm optimization with extreme learning machine (ELM) for daily reference evapotranspiration ($ET_0$) prediction from limited climatic data, *Zhu et al. (2020)* explored the ability of artificial neural networks (ANN), random forests (RF) and other empirical algorithms in estimating daily $ET_0$. The results indicated that the machine learning models outperformed the corresponding empirical algorithms. Machine learning algorithms are available for SSC inversion, and yet the accuracy of each algorithm using Sentinel-2 data to estimate SSC needs more in-depth comparison.

SSC evaluation at different root depths in the vegetated soil and the algorithm accuracy in SSC estimation via Sentinel-2 data both demands further research. Therefore, this study used the Sentinel-2 images of the Jiefangzha Irrigation Area (JIA) in the vegetated area to construct the set of independent variables (IVs, including 12 bands and 32 spectral indices). Next, the optimal combinations of IVs at five depths (0~20 cm, 20~40 cm, 40~60 cm, 0~40 cm, and 0~60 cm) were obtained using full subset selection. Finally, four machine learning algorithms, back propagation neural network (BPNN), support vector machine (SVM), extreme learning machine (ELM) and random forest (RF), were used to construct inversion models (models for SSC estimation via satellite data) and evaluate the sensitivity of Sentinel-2 to SSC at different depths and the inversion capability of the models.

## MATERIALS & METHODS

### Study area

The study area, JIA, is located in the northwest of Hetao Irrigation District (HID), Inner Mongolia, China, between 106°34′~107°34′E, and 40°26′~41°13′N, which is the same as that of *Qiu et al. (2019)*. JIA, an oblique triangular area about 86 km long and 81 km wide, is the second largest irrigation area in HID. With an altitude of about 1,030 m to 1,046 m, this plateau is high in the southwest, low in the northeast, and relatively flat on the whole. It is located in an arid and semiarid area with a temperate continental climate. The annual mean temperature, annual precipitation and evaporation are 4~6 mm, 66.3~200 mm and 1,920~3,450 mm, respectively. Moreover, the annual precipitation distribution is uneven (the precipitation in summer accounts for about 70% of the year). JIA is 2156.7 km$^2$ in total, about 57.5% of which is irrigatable. The crop planting structure is complex, and the crops mainly include corn, wheat, sunflower and so on. The local climate and hydrological conditions determine the dependence of the crop growth in this area mainly on the irrigation from the Yellow River. The annual water diversion in JIA is about 1.2 billion m$^3$, and the land is mainly irrigated in summer and autumn. The water diversion in October accounts for about 30% of the annual amount. Figure 1A shows the specific geographical location of the study area.

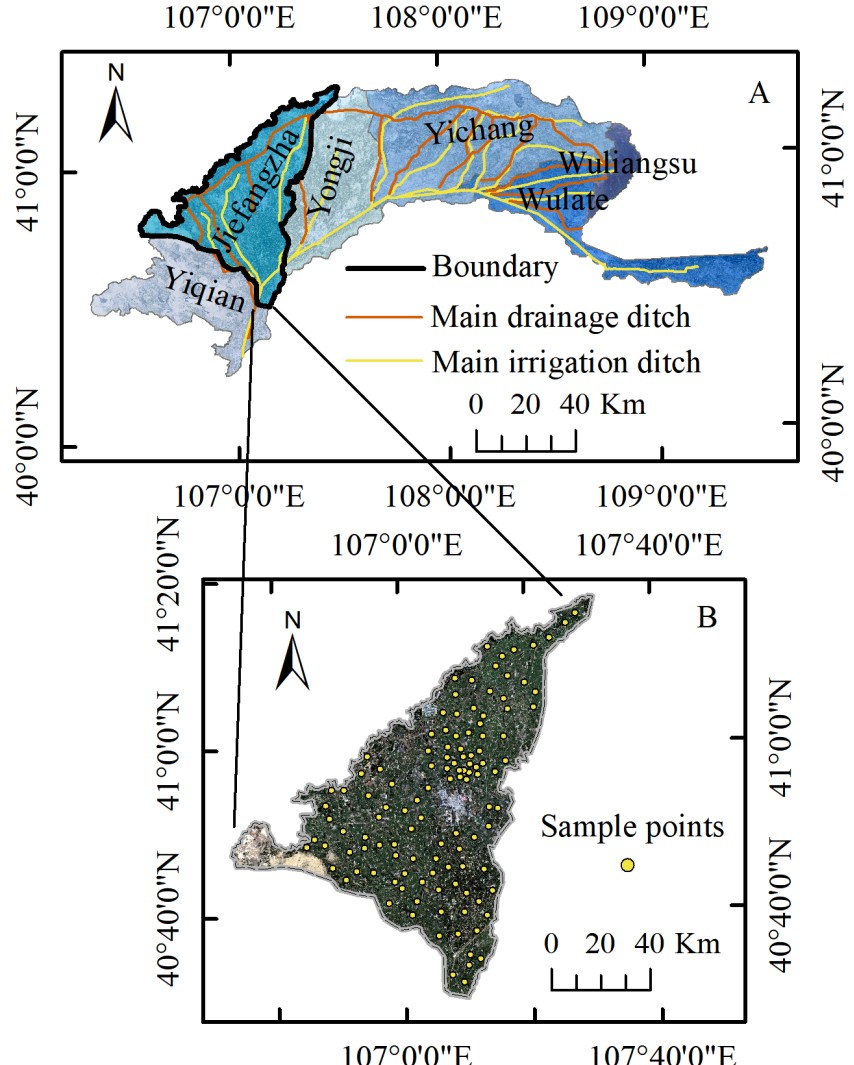

**Figure 1** **Study area and sample points location.** (A) Location of JIA in HID. (B) Sample points distribution.

## Soil sample collection and analysis

The Hetao irrigation district administration gave field permit approval to us (No. 2017YFC0403302). Among the main crops in JIA, sunflower, corn and zucchini are usually harvested around September 20. Therefore, August 25~30, 2019, when the crops were flourishing and the vegetation information was abundant, was chosen as the study period. We collected 117 soil samples (83 covered with sunflowers, 10 cabbages, 10 corns, 2 wheats, 2 vegetables, and 10 bare soil) (Fig. 1B) when the types of underlying surface, salinization degree, and evenness of point distribution were taken into consideration. According to the root depth of the crops and related research results (*Qiao, 2005*; *Zhang et al., 2019b*), we selected 0~20 cm 20~40 cm and 40~60 cm as the sampling depth. We adopted the five-point sampling method so that each of the 117 soil samples was the mixed

soil from one center and four corner points in a cell (0.5 m × 0.5 m). The GPS data and environment information of the center point were recorded during the sampling.

### Measurement of SSC

The soil samples were first dried, ground, and then screened with a 2.0-mm sieve to remove the small stones and wood pieces. The processed samples were mixed with water to make soil solution (the ratio of soil to water is 1:5). Eight hours after the solution was prepared, its electrical conductivity ($EC_{1:5}$, ds/cm) was measured by conductivity meter (DDS-307A; Shanghai Youke Instrument Branch, Shanghai, China), and then the SSC (%) was calculated by the empirical formula (1) , which was developed by *Huang et al. (2018)* in their research in HID.

$$SSC = (0.2882 \times EC_{1:5}) + 0.0183. \tag{1}$$

### Statistical characteristics of SSC

We selected five depths (0~20 cm, 20~40 cm, 40~60 cm, 0~40 cm and 0~60 cm) in this study, and the SSC of 0~40 cm was the mean value of 0~20 cm and 20~40 cm and that of 0~60 cm was the mean value of 0~20 cm, 20~40 cm and 40~60 cm. The sample points were sequenced according to the SSC, and then one of every three samples were selected as the validation dataset so that the ranges of calibration and validation datasets were consistent and evenly distributed. The statistical characteristics of SSC are shown in Table 1.

## Sentinel-2 image data

Sentinel-2 consists of two satellites (Sentinel-2A and 2B), and both provide multi-scale (10m, 20m and 60m) remote sensing images via MultiSpectral Instrument (MSI). It has a 5-day revisiting period when the two satellites are combined. It has 13 bands (440 nm~2200 nm), including visible light, near infrared and short-wave infrared bands. Three of the bands can obtain the red-edge region of vegetation spectrum (670 nm~760 nm), which can provide more effective data for vegetation growth monitoring. The remote sensing image data for this study were taken from Sentinel-2A, and its parameters of each band are shown in Table 2.

The satellite images in this study were obtained from the website (https://scihub.copernicus.eu/) of the European Space Agency (ESA), on August 28, 2019, which was basically synchronous with the sampling time and there was no cloud in the study area. The Sentinel-2 data used in this study were Level-2A products (Bottom-Of-Atmosphere reflectance images), which were produced by the plug-in Sen2cor (Level-2A product can also be obtained directly from ESA since December 2018). Then all the bands were resampled to 10m via the S2 Resampling Processor in the software SNAP (the bicubic method was used for resampling). As band B10 was not available when the images were processed to Level-2A, the other 12 bands were used in this study.

**Table 1  Statistical characteristics of SSC.** SSC is soil salt content, SD is standard deviation, CV is coefficient of variation, and the Number refers to the number of sampling points.

| Depth (cm) | Dataset | Number | Min. (%) | Max. (%) | Average (%) | SD (%) | CV (%) |
|---|---|---|---|---|---|---|---|
| | Total | 117 | 0.069 | 1.66 | 0.30 | 0.26 | 88.4 |
| 0~20 | Calibration | 78 | 0.069 | 1.66 | 0.30 | 0.28 | 92.2 |
| | Validation | 39 | 0.076 | 1.26 | 0.29 | 0.23 | 80.6 |
| | Total | 117 | 0.070 | 1.40 | 0.28 | 0.23 | 82.6 |
| 20~40 | Calibration | 78 | 0.070 | 1.40 | 0.27 | 0.22 | 82.1 |
| | Validation | 39 | 0.082 | 1.27 | 0.29 | 0.24 | 84.3 |
| | Total | 117 | 0.082 | 1.48 | 0.28 | 0.22 | 79.7 |
| 40~60 | Calibration | 78 | 0.083 | 1.48 | 0.28 | 0.23 | 81.6 |
| | Validation | 39 | 0.082 | 1.15 | 0.28 | 0.22 | 76.6 |
| | Total | 117 | 0.070 | 1.40 | 0.29 | 0.24 | 83.1 |
| 0~40 | Calibration | 78 | 0.070 | 1.40 | 0.29 | 0.24 | 84.5 |
| | Validation | 39 | 0.079 | 1.26 | 0.29 | 0.24 | 81.4 |
| | Total | 117 | 0.075 | 1.43 | 0.28 | 0.23 | 80.0 |
| 0~60 | Calibration | 78 | 0.075 | 1.43 | 0.28 | 0.23 | 81.1 |
| | Validation | 39 | 0.085 | 1.22 | 0.29 | 0.23 | 78.9 |

**Table 2  Sentinel-2A band parameters.**

| Band | Band center (nm) | Spatial resolution (m) |
|---|---|---|
| B1 (Coastal aerosol) | 443.9 | 60 |
| B2 (Blue) | 496.9 | 10 |
| B3 (Green) | 560.0 | 10 |
| B4 (Red) | 664.5 | 10 |
| B5 (Red-edge 1) | 703.9 | 20 |
| B6 (Red-edge 2) | 740.2 | 20 |
| B7 (Red-edge 3) | 782.5 | 20 |
| B8 (NIR) | 835.1 | 10 |
| B8A (Narrow NIR) | 864.8 | 20 |
| B9 (Water Vapor) | 945.0 | 60 |
| B10 (Cirrus) | 1373.5 | 60 |
| B11 (SWIR1) | 1613.7 | 20 |
| B12 (SWIR2) | 2202.4 | 20 |

## Extraction and selection of spectral index
### Selection of spectral index
In this study, we selected 32 widely used spectral indices, including salinity index, vegetation index and drought index. The indices and the relevant formulae are shown in Table 3.

### Soil line fitting
As has been shown in the Nir-Red scatterplot of several studies, when the horizontal and vertical coordinates are the red and Nir bands, respectively, a series of corresponding points of the digital number values of the red and Nir infrared wavelengths of the bare soil
**Table 3 Spectral indices.** M is the slope of soil line, I is the intercept of soil line, A is the PVI maximum point. The fitting result of soil line is shown in Fig. 2.

| Spectral index | formula | Reference |
|---|---|---|
| Salinity Index (SI) | $\sqrt{B2 \times B4}$ | |
| Salinity Index-2 (SI2) | $\sqrt{B3^2 + B4^2 + B8A^2}$ | |
| Salinity Index 3 (SI3) | $\sqrt{B3^2 + B4^2}$ | |
| Salinity Index (S1) | $B2/B4$ | |
| Salinity Index (S2) | $(B2 - B4)/(B2 + B4)$ | |
| Salinity Index (S3) | $(B3 \times B4)/B2$ | *Allbed, Kumar & Aldakheel (2014)* |
| Salinity Index (S5) | $(B2 \times B4)/B3$ | |
| Salinity Index (S6) | $(B4 \times B8A)/B3$ | |
| Salinity Index-T (SI-T) | $(B4/B8A)/100$ | |
| Normalized Difference salinity Index (NDSI) | $(B4 - B8A)/(B4 + B8A)$ | |
| Brightness Index (BI) | $\sqrt{B4^2 + B8A^2}$ | |
| Canopy Response Salinity Index (CRSI) | $\sqrt{\dfrac{B8A \times B4 - B3 \times B2}{B8A \times B4 + B3 \times B2}}$ | *Scudiero, Skaggs & Corwin (2015)* |
| Intensity index 1 (Int1) | $(B3 + B4)/2$ | *Triki Fourati et al. (2015)* |
| Intensity index 2 (Int2) | $(B3 + B4 + B8A)/2$ | |
| Normalized Vegetation Index (NDVI) | $(B8A - B4)/(B8A + B4)$ | *Cho, Beon & Jeong (2018)* |
| Enhanced Vegetation Index (EVI) | $\dfrac{2.5 \times (B8A - B4)}{(B8A + 6 \times B4 - 7.5 \times B2 + 1)}$ | *Qiu et al. (2019)* |
| Red Edge Position Index (S2$_{REP}$) | $705 + \dfrac{35 \times ((B4 + B7)/2 - B5)}{(B6 - B5)}$ | |
| Normalized Difference Vegetation Index red-edge 1 (NDVIrel) | $(B8A - B5)/(B8A + B5)$ | |
| Normalized Difference Vegetation Index red-edge 2 (NDVIre2) | $(B8A - B6)/(B5 + B6)$ | |
| Normalized Difference red-edge 1 (NDre1) | $(B6 - B5)/(B6 + B5)$ | *Gu (2019)* |
| Normalized Difference red-edge 2 (NDre2) | $(B7 - B5)/(B7 + B5)$ | |
| Triangular Chlorophyll Index red-edge 1 (TCIrel) | $1.2 \times (B5 - B3) - 1.5 \times (B4 - B3) \times \sqrt{B5/B4}$ | |
| Modified soil-adjusted Vegetation Index (MSAVI) | $\dfrac{2 \times B4 + 1 - \sqrt{(2 \times B8A + 1)^2 - 8 \times (B8A - B4)}}{2}$ | |
| Normalized Difference Drought Index (NDDI) | $(NDVI - NDWI)/(NDVI + NDWI)$ | *Wang, Li & Li (2019)* |
| Soil Moisture Monitoring Index (SMMI) | $\sqrt{B8A^2 + B11^2}/\sqrt{2}$ | |
| Perpendicular Vegetation Index (PVI) | $|B8A - M \times B4 - I|/\sqrt{M^2 + 1}$ | |
| Perpendicular Drought Index (PDI) | $(B4 + M \times B8A)/\sqrt{M^2 + 1}$ | *Wu et al. (2014a)* |
| Vegetation Adjusted Perpendicular Drought Index (VAPDI) | $PDI(A) - \dfrac{|PDI(A) - PDI(X)| \times PVI(A)}{PVI(A) - PVI(X)}$ | |
| Normalized Shortwave-infrared Difference SM Index 3 (NSDSI3) | $(B11 - B12)/(B11 + B12)$ | *Yue et al. (2019)* |
| Normalized Multiband Drought Index (NMDI) | $\dfrac{B8A - (B11 - B12)}{B8A + (B11 + B12)}$ | |
| Normalized Difference Water Index (NDWI) | $(B8A - B11)/(B8A + B11)$ | *Khaled (2017)* |
| Visible and Shortwave Drought Index (VSDI) | $1 - (B12 + B4 - 2 \times B2)$ | |

approximate to fit into a straight line, which is called soil line (*Wu et al., 2014a*). Three spectral indices (PVI, PDI and VAPDI) used in this study were based on the concept of soil line. Eight hundred pure bare pixels (NDVI <0.1) were identified by visual interpretation

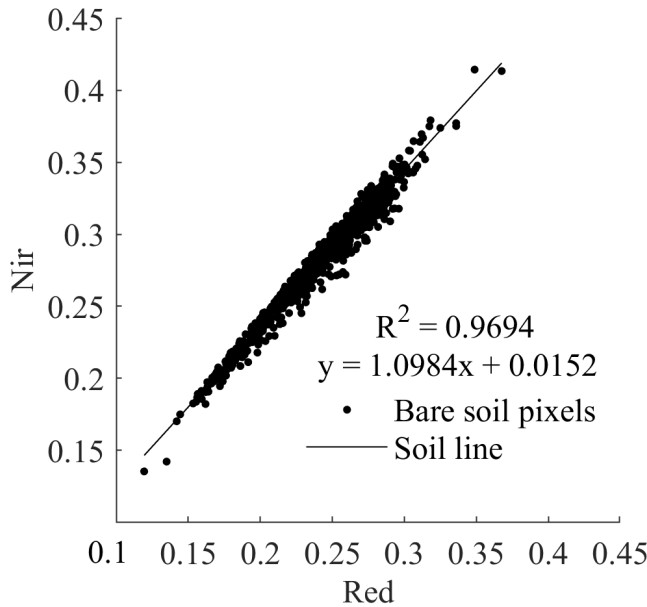

**Figure 2  Soil line fitting diagram of JIA.**

using conventional soil line extraction for soil line fitting. $R^2$, $M$, and $I$ were 0.9694, 1.0984, and 0.0152, respectively (Fig. 2).

### Full subset selection

According to the least squares method, the full subset selection is a method to select the optimal combinations by traversing all possible combinations of the full IV set (the IV set includes 12 bands and 32 indices). The calculation of the full subset selection takes the following steps: When the number of IVs is P, $P$ models can be built according to $K$, the number of IVs input into the model ($1 \leq K \leq P$, $K$ is integer), and there are $C_P^K$ combinations of IVs for each model (*Zhang et al., 2019b*). Therefore, based on the calibration dataset, the optimal combination of the IVs for each model was selected according to the maximum of the adjusted coefficient of determination ($R^2_{adj}$). Afterwards, based on the $R^2_{adj}$, root mean square error (RMSE), mean absolute error (MAE), Akaike information criterion (AIC) and Bayesian information criterion (BIC), the optimal model from P models at each depth was selected on the ground of the validation dataset. Considering the computational magnitude problem of full subset selection, the value of $K$ was taken from 2 to 6.

Among the five criteria, $R^2_{adj}$ can improve the accuracy of the comparison between the models with different numbers of IVs and samples. As the number of IVs in the model increases, $R^2_{adj}$ will not necessarily increase (*Srivastava, Srivastava & Ullah, 1995*), which mitigates the difference among the coefficient of determination caused by the number of IVs. RMSE and MAE are indicators to evaluate the model inversion error; AIC and BIC measure the goodness of model fit. Smaller values of AIC and BIC mean the model can explain the dependent variable (DV) with fewer IVs (*Atkinson et al., 2012*). The equations

are shown in Eqs. (2)–(6)

$$R^2_{adj} = 1 - (1 - \frac{\sum_{i=1}^{n}(\hat{y}_i - \overline{y})^2}{\sum_{i=1}^{n}(y_i - \overline{y})^2}) \frac{(n-1)}{(n-k)} \tag{2}$$

$$RMSE = \sqrt{\frac{\sum_{i=1}^{n}(\hat{y}_i - y_i)^2}{n}} \tag{3}$$

$$MAE = \frac{1}{n}\sum_{i=1}^{n}|\hat{y}_i - y_i| \tag{4}$$

$$AIC = 2k + n[ln(RSS)] \tag{5}$$

$$BIC = n[ln(\hat{\sigma}^2)] + k \times ln(n) \tag{6}$$

where $\hat{y}_i$, $y_i$ and $\overline{y}$ are the predicted, measured, and the average of measured values of the model, respectively; $n$ is the number of samples; $k$ is the number of free parameters in the model; RSS is the squared sum of the residuals between the measured and predicted data; $\hat{\sigma}^2$ is the error variance.

## Construction of machine learning models

Four machine learning algorithms, BPNN, SVM, ELM and RF were selected for SSC estimation. Figure 3 is the flowchart of the proposed methodology of SSC estimation in this study.

### *BPNN Model*

BPNN algorithm, proposed by *Rumelhart, Hinton & Williams (1986)*, has a strong nonlinear mapping capability and can adjust the internal parameters of the system according to the error between the output and actual value via the error back propagation algorithm. Topologically, the BPNN model consists of three layers: the input, hidden, and output layers (Fig. 4) (*Wang et al., 2018*). After extensive pre-testing, the BPNN model in this study used the optimal combination of IVs as the input layer, SSC as the output layer, and the number of hidden layers was set as 2. The transfer functions of the input and output layers were linear, and the hidden layers were tangent-S. The target error and network learning rate were 0. $65 \times 10^{-3}$ and 0.05, respectively. In order to eliminate the effect of different dimensions on data analysis, the input layer and output layer data were normalized (so were the other three models). MATLAB was used to build the BPNN model (so were the other three models). Details of BPNN can be found in *Xiao et al. (2020)* and *Chen et al. (2015)*.

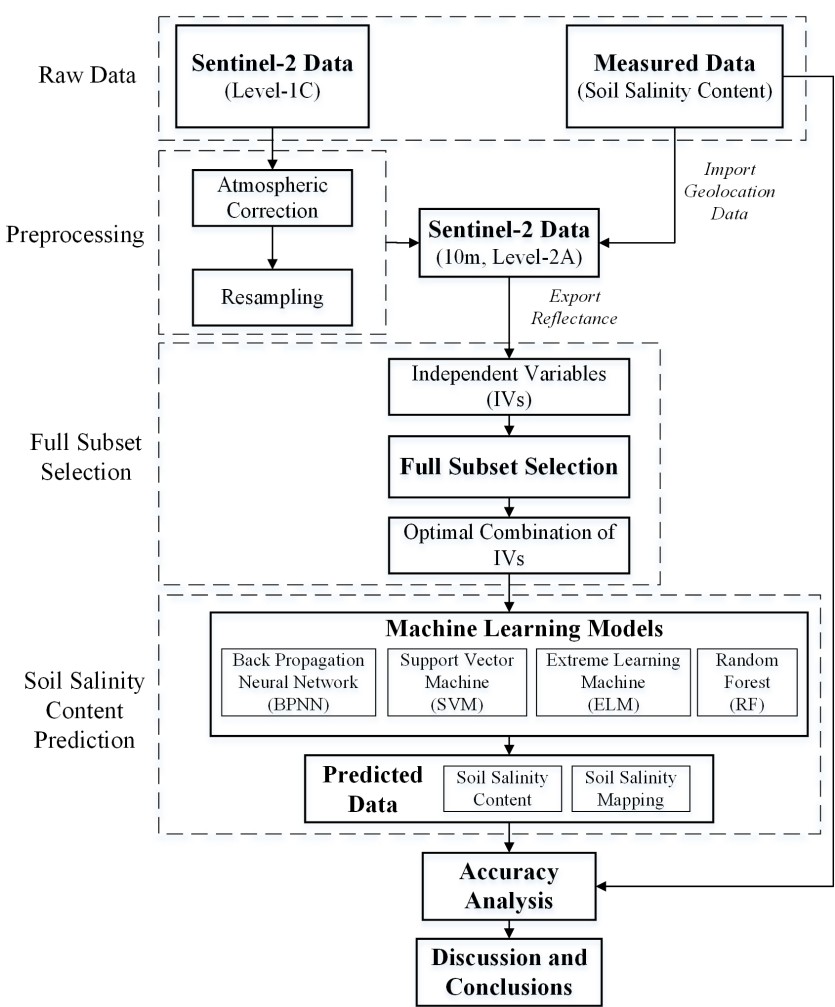

**Figure 3** Flowchart of the proposed methodology for SCC estimation.

### SVM Model

SVM is a machine learning algorithm based on the principle of structural risk minimization. It focuses on transforming the input data into a high-dimensional feature space using nonlinear transformations for classification and regression (*Chen et al., 2015*). This algorithm enjoys such advantages as avoiding discrete values, mitigating overlearning and reducing computation. This study adopted the widely used radial basis kernel (RBF) (*Zhang et al., 2019b*) as the kernel function of SVM. The penalty parameter ($C$) and the nuclear parameter ($g$) of the RBF have a great effect on the model stability, so a grid-searching technique was used to find the best parameters of $C$ and $g$ (at 0∼20 cm 20∼40 cm 40∼60 cm ∼40 cm and 0∼60 cm, the $C$ was 724, 1024, 1024, 1024 and 1024, respectively, the $g$ was 0.0313, 0.0028, 0.0039, 0.0156 and 0.01, respectively).

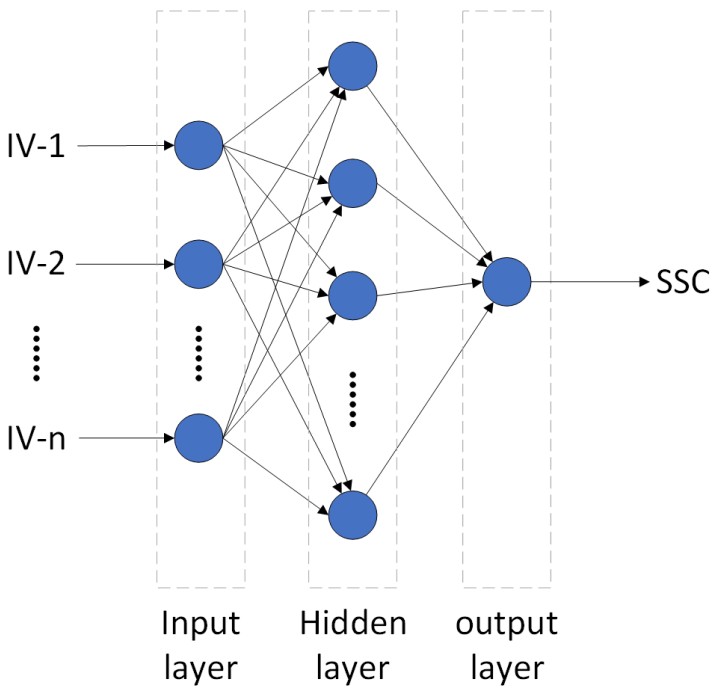

**Figure 4** **The topological structure of BPNN and ELM.**

### ELM Model

ELM is a machine learning algorithm proposed by *Huang, Zhu & Siew (2006)*. It is the same as BPNN in structure, a traditional three-layer neural network (Fig. 4). Its main difference is that the execution does not require adjustments to the input weights of the network or the biases of the hidden elements. Therefore, ELM can reduce the influence of such subjective factors as choice of parameters, and speed up computation while ensure accuracy (*Ahila, Sadasivam & Manimala, 2015*; *Prasad et al., 2019*). After extensive pre-testing, this study adopted sigmoid as the activation function, and the number of hidden nodes was set as 6. Detailed principles of ELM can be found in *Zhu et al. (2020)*.

### RF Model

RF is a machine learning algorithm proposed by *Breiman (2001)*. Based on multiple decision tree theory, this algorithm can be used for classification and regression. RF uses the bootstrap method to extract training sets from the input data, and randomly generates variables to build decision tree models. Thus, the decision made by a random forest model is based on the ensemble of decisions made by numerous decision trees (Fig. 5) (*Zhou et al., 2020*; *Du et al., 2015*). After extensive pre-testing, the minimum number of observations per tree leaf (minleaf) and the number of decision trees (ntree) were set as 5 and 10, respectively (the two parameters were determined by the out-of-bag errors and training set cross-validation) (*Zhang et al., 2019a*).

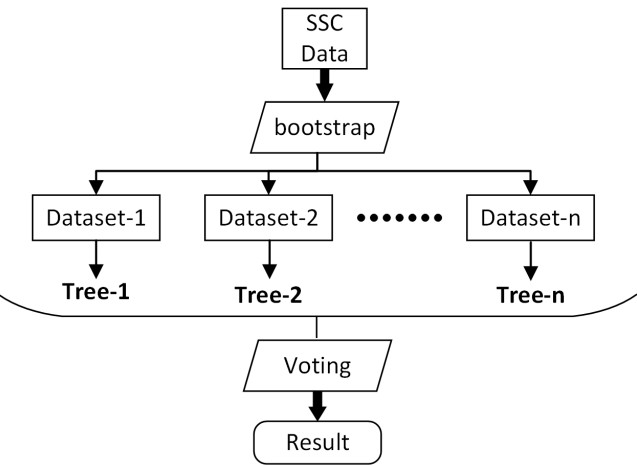

**Figure 5  The structure of RF.**

*Model accuracy evaluation*

$R^2_{adj}$, RMSE, and MAE were used to evaluate the inversion of calibration and validation model. Among the three criteria, $R^2_{adj}$ can avoid the errors caused by the different number of IVs in each model. The closer its value is to 1, the better fit the model has. The smaller the RMSE and MAE are, the smaller the deviation between the predicted and measured values are. The equations are shown in Eqs. (2)–(4).

## RESULTS AND ANALYSIS

### Analysis of correlation between SSC and IVs

We selected 12 bands of Sentinel-2 and 32 spectral indices to form the IV set for SSC inversion. Based on the calibration dataset, the correlation between the IVs and SSC was analyzed, as is shown in Fig. 6.

The significance level between the IVs and SSC was tested according to the significance testing table of correlation coefficient. When the degree of freedom was 78 and the absolute value of the correlation coefficient ($R$) was greater than 0.221, the significance level reached 0.05, and when R was greater than 0.288, the significance level reached 0.01. At all five depths (0∼20 cm, 20∼40 cm, 40∼60 cm, 0∼40 cm and 0∼60 cm), the significance level between eight IVs (B6, SI2, S1, S2, S6, BI, SMMI, and PDI) and SSC failed to reach 0.05, that between six IVs (B7, B8, B8A, Int2, NSDSI3, and VSDI) and SSC reached the 0.05, and that between the remaining 30 IVs and SSC reached 0.01.

### Optimal combination of IVs based on full subset selection

In order to identify the optimal combinations of IVs, we took the SSC as the DV, and the least squares as the method for data fitting on the basis of the calibration dataset. Then we obtained the optimal combination when the number of IVs was 2 to 6 at the five depths (0∼20 cm, 20∼40 cm, 40∼60 cm, 0∼40 cm and 0∼60 cm) and calculated the $R^2_{adj}$, RMSE, MAE, AIC and BIC based on the validation dataset (Table 4).

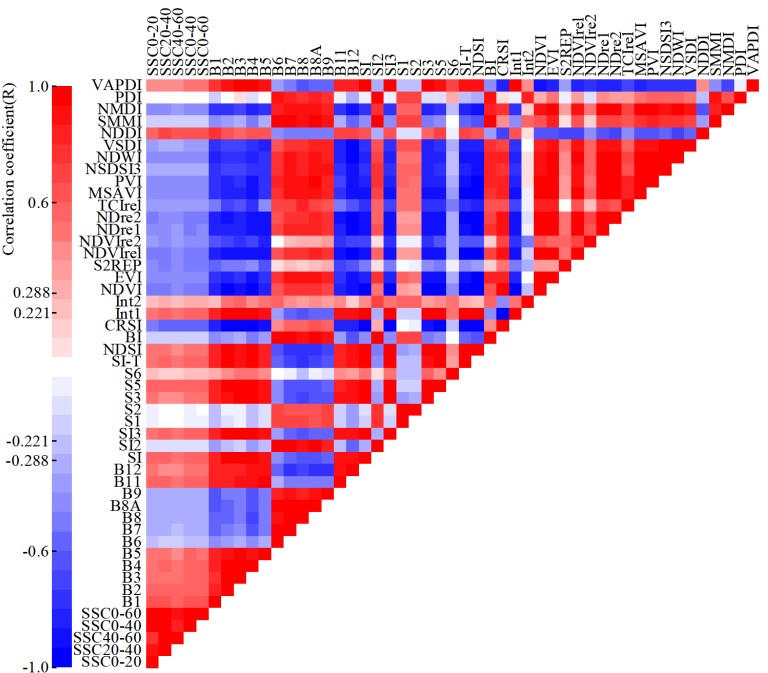

**Figure 6** Heatmap of Pearson correlation coefficient between IVs and SSC.

$R^2_{adj}$ demonstrated a tendency of rising and then falling when the number of IVs increased at most depths, and this tendency was especially obvious at the depths of 20∼40 cm, 40∼60 cm and 0∼60 cm. This indicated that the model fitting became better when the number of IVs was on the rise, but too many IVs would complicate the model, leading to some overfitting. RMSE and MAE displayed the opposite tendency of $R^2_{adj}$: they first fell and then rose. This indicated that a local optimal solution might be generated and the validation model error might be increased when there were too many IVs. The tendency of AIC and BIC was almost the same, when the number of IVs increased, there was a first-falling-and-then-rising tendency at 20∼40 cm, 40∼60 cm and 0∼60 cm, and a tendency of increase at 0∼20 cm and 0∼40 cm.

At 0∼20 cm, when the numbers of IVs were 2 (IV-2) and 6 (IV-6), the $R^2_{adj}$ was the highest (both were 0.44), and those of the RMSE and MAE were relatively close. However, the AIC of IV-2 was half of that of IV-6, suggesting that IV-2 was able to explain the DV with fewer IVs. Therefore, B1 and NSDSI3 were the optimal combination of IVs at 0∼20 cm. At 20∼40 cm, the $R^2_{adj}$ of the IV-3 was 0.58, which was significantly higher than that of the other combinations. The remaining criteria of IV-3 were the lowest, indicating a small error and a high goodness of fit. Therefore, S5, SI-T and NDDI were the optimal combination of IVs at 20∼40 cm. The tendency at 40∼60 cm was the same as that at 20∼40 cm. IV-3 had the highest $R^2_{adj}$, relatively lower errors and higher goodness of fit at the same time. Therefore, S6, VSDI and NDDI were the optimal combination of IVs at 40∼60 cm. At 0∼40 cm, the $R^2_{adj}$ of IV-2 to IV-4 were all above 0.45 but the AIC and BIC of

**Table 4  Optimal combination of IVs based on full subset selection.**

| Depth (cm) | IV | Optimal combinations | $R^2_{adj}$ | RMSE (%) | MAE (%) | AIC | BIC |
|---|---|---|---|---|---|---|---|
| | **2** | **B1**`NSDSI3**** | 0.44 | 0.17 | 0.13 | 8.8 | −130 |
| | 3 | B1**`NSDSI3**`NMDI** | 0.43 | 0.17 | 0.13 | 11.3 | −126 |
| 0~20 | 4 | B1**`B11**`NSDSI3**`SMMI | 0.40 | 0.18 | 0.13 | 15.0 | −120 |
| | 5 | B1**`B11**`B12**`S2$_{REP}$**`PVI** | 0.33 | 0.19 | 0.14 | 21.5 | −112 |
| | 6 | B5**`B8A**`B12**`NDre2**`NSDSI3***`SMMI | 0.44 | 0.17 | 0.13 | 16.5 | −115 |
| | 2 | S6`NDDI** | 0.44 | 0.18 | 0.14 | 12.4 | −126 |
| | 3 | **S5**`SI-T**`NDDI**** | 0.58 | 0.15 | 0.12 | 2.7 | −134 |
| 20~40 | 4 | SI3**`NDWI**`NDDI**`NMDI** | 0.50 | 0.17 | 0.13 | 12.1 | −123 |
| | 5 | NDVI**`TCIrel**`NDWI**`NDDI**`NMDI** | 0.42 | 0.18 | 0.14 | 19.9 | −114 |
| | 6 | NDVI**`S2$_{REP}$**`TCIrel**`NDWI**`NDDI**`NMDI** | 0.35 | 0.19 | 0.14 | 26.4 | −105 |
| | 2 | S6`NDDI** | 0.38 | 0.16 | 0.12 | 6.2 | −132 |
| | 3 | **S6`VSDI*`NDDI**** | 0.53 | 0.14 | 0.11 | −2.5 | −139 |
| 40~60 | 4 | B2**`NDWI**`NDDI**`NMDI** | 0.45 | 0.16 | 0.12 | 5.9 | −129 |
| | 5 | NDVI**`NDVIre2**`NDWI**`NDDI**`NMDI** | 0.38 | 0.16 | 0.12 | 12.2 | −121 |
| | 6 | S3**`NDVI**`NDVIre2**`NDWI**`NDDI**`NMDI** | 0.34 | 0.17 | 0.13 | 16.4 | −116 |
| | 2 | **B1**` NSDSI3**** | 0.45 | 0.17 | 0.13 | 6.5 | −132 |
| | 3 | B1**`NSDSI3**`NMDI** | 0.45 | 0.17 | 0.13 | 8.7 | −128 |
| 0~40 | 4 | Int2`NDWI**`NDDI**`NMDI** | 0.48 | 0.16 | 0.12 | 8.8 | −126 |
| | 5 | B1**`B11**`B12**`S2$_{REP}$**`PVI** | 0.29 | 0.19 | 0.14 | 22.5 | −111 |
| | 6 | B11**`B12**`NDVI**`NDWI**`NDDI**`PDI | 0.42 | 0.17 | 0.13 | 16.8 | −115 |
| | 2 | S6*`NDDI** | 0.37 | 0.17 | 0.13 | 7.9 | −131 |
| | 3 | **B1**`PVI**`NSDSI3**** | 0.46 | 0.16 | 0.12 | 3.6 | −133 |
| 0~60 | 4 | SI3**`NDWI**`NDDI**`NMDI** | 0.46 | 0.16 | 0.12 | 6.2 | −129 |
| | 5 | B1**`NDVI**`NDWI**`NDDI**`NMDI** | 0.49 | 0.15 | 0.12 | 5.3 | −128 |
| | 6 | SI-T**`BI`NDWI**`NDDI**`SMMI`NMDI** | 0.44 | 0.16 | 0.12 | 11.1 | −121 |

**Notes.**
*Significance level of 0.05.
**Significance level of 0.01.
Bold text represents the optimal combination out of the optimal combinations of IVs at each depth.

IV-2 were the lowest among the three. IV-2 needed the smallest number of IVs to achieve a better fit. Therefore, B1 and NSDSI3 were the optimal combination of IVs at 0~40 cm. At 0~60 cm, the $R^2_{adj}$ of IV-3 to IV-6 were all above 0.44, and the AIC and BIC of IV-3 were the lowest (3.6 and −133, respectively). The RMSE and MAE were relatively low. Therefore, B1, PVI and NSDSI3 were the optimal combination of IVs at 0~60 cm.

## Model calibrations and validations
### Analysis of BPNN model

The optimal combination (after the full subset selection) of IVs at each depth was input into the BPNN model for training to obtain the SSC inversion model. The accuracy of calibration and validation models are shown in Table 5.

The model performance was optimal at 20~40 cm, with the $R^2_{adj}$ of around 0.6 for both calibration and validation models, indicating a good fitting and generalization performance.

**Table 5   SSC inversion based on BPNN model.**

| Depth (cm) | Calibration | | | Validation | | |
|---|---|---|---|---|---|---|
| | $R^2_{adj}$ | RMSE (%) | MAE (%) | $R^2_{adj}$ | RMSE (%) | MAE (%) |
| 0∼20 | 0.48 | 0.20 | 0.13 | 0.46 | 0.17 | 0.13 |
| 20∼40 | 0.59 | 0.14 | 0.11 | 0.61 | 0.15 | 0.11 |
| 40∼60 | 0.56 | 0.15 | 0.12 | 0.52 | 0.14 | 0.11 |
| 0∼40 | 0.53 | 0.17 | 0.12 | 0.50 | 0.16 | 0.12 |
| 0∼60 | 0.52 | 0.16 | 0.12 | 0.51 | 0.15 | 0.12 |

Its RMSE (0.15%) and MAE (0.11%) were relatively low, indicating their good control of inversion errors. The model had the worst performance at 0∼20 cm, with the lowest $R^2_{adj}$ (below 0.48) of the calibration and verification model, and the RMSE (both above 0.17%) and MAE (0.13%) were the highest at the five depths. At 40∼60 cm, the model performance was second only to 20∼40 cm, with a relatively good fitting and low inversion error. The performance of the model at 0∼40 cm and 0∼60 cm was similar. The $R^2_{adj}$ and MAE were around 0.51 and 0.12%, respectively, but the RMSE at 0∼40 cm was greater than that at 0∼60 cm, suggesting that some samples had some relatively big errors.

Overall, the BPNN model worked best at 20∼40 cm, followed by 40∼60 cm, and worked worst at 0∼20 cm. The other two depths had better similar results.

### Analysis of SVM model

The optimal combination (after the full subset selection) of IVs at each depth was input into the SVM model for training to obtain the SSC inversion model. The accuracy of calibration and validation models are shown in Table 6.

The model performance was optimal at 40∼60 cm, with $R^2_{adj}$ of 0.58 and 0.55 for the calibration and validation models, respectively, and its RMSE and MAE were the lowest. The performance of the model at 20∼40 cm and 0∼40 cm was similar, second only to 40∼60 cm. Comparatively, 20∼40 cm was slightly better in fitting of validation model. At 0–20 cm, the model performance was still the worst. Its RMSE (above 0.16%) and MAE (above 0.12%) indicated a bad control of inversion error. But its $R^2_{adj}$ was around 0.5, the model at this depth still had a good fitting. At 0∼60 cm, the $R^2_{adj}$ of the calibration and verification models were 0.59 and 0.52, respectively, showing a slight overfitting.

Overall, the SVM model worked best at 40∼60 cm, followed by 20∼40 cm and 0∼40 cm. The worst performance of the model was found at 0∼20 cm.

### Analysis of ELM model

The optimal combination (after the full subset selection) of IVs at each depth was input into the ELM model for training to obtain the SSC inversion model. The accuracy of calibration and validation models are shown in Table 7.

The model performance was optimal at 20∼40 cm, with the highest $R^2_{adj}$ (above 0.6) and lowest inversion error. The model was still the worst at 0∼20 cm, mainly because the inversion error was the largest. But it still had a good fitting ($R^2_{adj} \approx 0.5$). The performance

**Table 6 SSC inversion based on SVM model.**

| Depth (cm) | Calibration | | | Validation | | |
|---|---|---|---|---|---|---|
| | $R^2_{adj}$ | RMSE (%) | MAE (%) | $R^2_{adj}$ | RMSE (%) | MAE (%) |
| 0~20 | 0.50 | 0.19 | 0.13 | 0.49 | 0.16 | 0.12 |
| 20~40 | 0.56 | 0.15 | 0.11 | 0.53 | 0.16 | 0.12 |
| 40~60 | 0.58 | 0.14 | 0.11 | 0.55 | 0.14 | 0.10 |
| 0~40 | 0.56 | 0.16 | 0.11 | 0.52 | 0.15 | 0.12 |
| 0~60 | 0.59 | 0.15 | 0.11 | 0.52 | 0.14 | 0.11 |

**Table 7 SSC inversion based on ELM model.**

| Depth (cm) | Calibration | | | Validation | | |
|---|---|---|---|---|---|---|
| | $R^2_{adj}$ | RMSE (%) | MAE (%) | $R^2_{adj}$ | RMSE (%) | MAE (%) |
| 0~20 | 0.51 | 0.19 | 0.13 | 0.49 | 0.16 | 0.13 |
| 20~40 | 0.60 | 0.14 | 0.10 | 0.62 | 0.14 | 0.11 |
| 40~60 | 0.58 | 0.14 | 0.11 | 0.52 | 0.14 | 0.11 |
| 0~40 | 0.58 | 0.16 | 0.11 | 0.55 | 0.15 | 0.12 |
| 0~60 | 0.52 | 0.16 | 0.12 | 0.51 | 0.15 | 0.11 |

of the model at 0~60 cm was slightly better than that at 0~20 cm. The model performance was satisfactory and similar at 40~60 cm and 0~40 cm.

In general, the ELM model worked best at 20~40 cm, but worst at 0~20 cm and 0~60 cm (0~60 cm was slightly better). This model had relatively good performance at the other two depths.

### Analysis of RF model

The optimal combination (after the full subset selection) of IVs at each depth was input into the RF model for training to obtain the SSC inversion model. The accuracy of calibration and validation models are shown in Table 8.

The RF model performed well at all five depths, with $R^2_{adj}$ all above 0.5. The model performance was still optimal at 20~40 cm. The $R^2_{adj}$ of the calibration and validation models were 0.68 and 0.63, respectively, which were significantly better than at the other depths. Its RMSE (below 0.14%) and MAE (below 0.11%) indicated that the inversion error was well controlled. At 0~40 cm, the model performance was second only to that at 20~40 cm. The result was worst at 0~60 cm. Its RMSE and MAE were around 0.15% and 0.12%, respectively, indicating a relatively high inversion error. However, its $R^2_{adj}$ also reached 0.53. The model performance at 0~20 cm was slightly better than that at 0~60 cm because the former's $R^2_{adj}$ of the calibration and validation models were higher (0.03 and 0.01 higher, respectively) than that of the latter. There was some overfitting at 40~60 cm.

In general, the RF model had a good performance at all depths, working best at 20~40 cm. The performance was relatively poor at 0~20 cm and 0~60 cm (0~20 cm was slightly better). The scatterplot of measured and predicted SSC of the four models at the best depth are shown in Fig. 7.

**Table 8   SSC inversion based on RF model.**

| Depth (cm) | Calibration | | | Validation | | |
|---|---|---|---|---|---|---|
| | $R^2_{adj}$ | RMSE (%) | MAE (%) | $R^2_{adj}$ | RMSE (%) | MAE (%) |
| 0~20 | 0.62 | 0.17 | 0.10 | 0.54 | 0.15 | 0.12 |
| 20~40 | 0.68 | 0.13 | 0.09 | 0.63 | 0.14 | 0.11 |
| 40~60 | 0.64 | 0.13 | 0.09 | 0.50 | 0.15 | 0.11 |
| 0~40 | 0.65 | 0.14 | 0.09 | 0.61 | 0.14 | 0.11 |
| 0~60 | 0.59 | 0.15 | 0.10 | 0.53 | 0.14 | 0.12 |

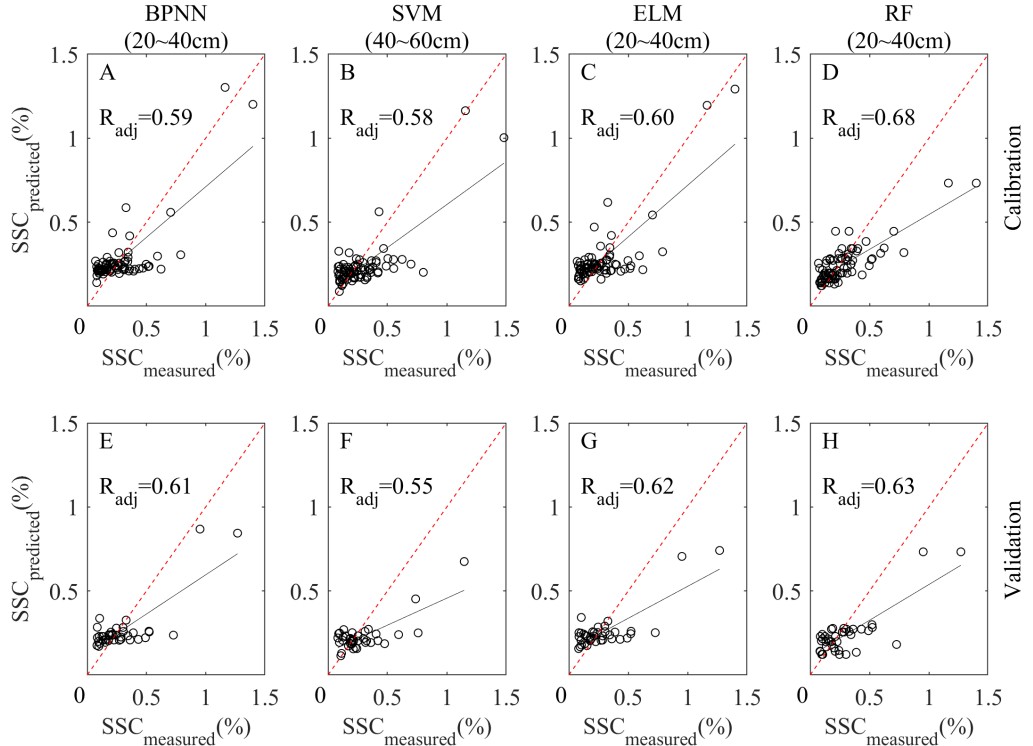

**Figure 7   Scatterplots of measured and predicted SSC of BPNN, SVM, ELM and RF models at the best depth.** (A–D) Calibration scatterplots of the four models, respectively. (E–H) Validation scatterplots of the four models, respectively.

## Evaluation of the overall inversion performance
### Evaluation of inversion depths

The inversion performance of each model at different depths has been discussed in detail (the Section of *Model calibrations and validations*). In this section, the sensitivity of Sentinel-2 to SSC at different depths was evaluated by analyzing the combined performance of all models at each depth (Fig. 8). At 20~40 cm, the $R^2_{adj}$ was significantly higher than that of other depths (Figs. 8A–8E), and each model was able to achieve a good fitting and generalization performance. At this depth, the RMSE and MAE of the calibration models were the lowest, and those of the validation models were also relatively low (Figs. 8F–8O). It

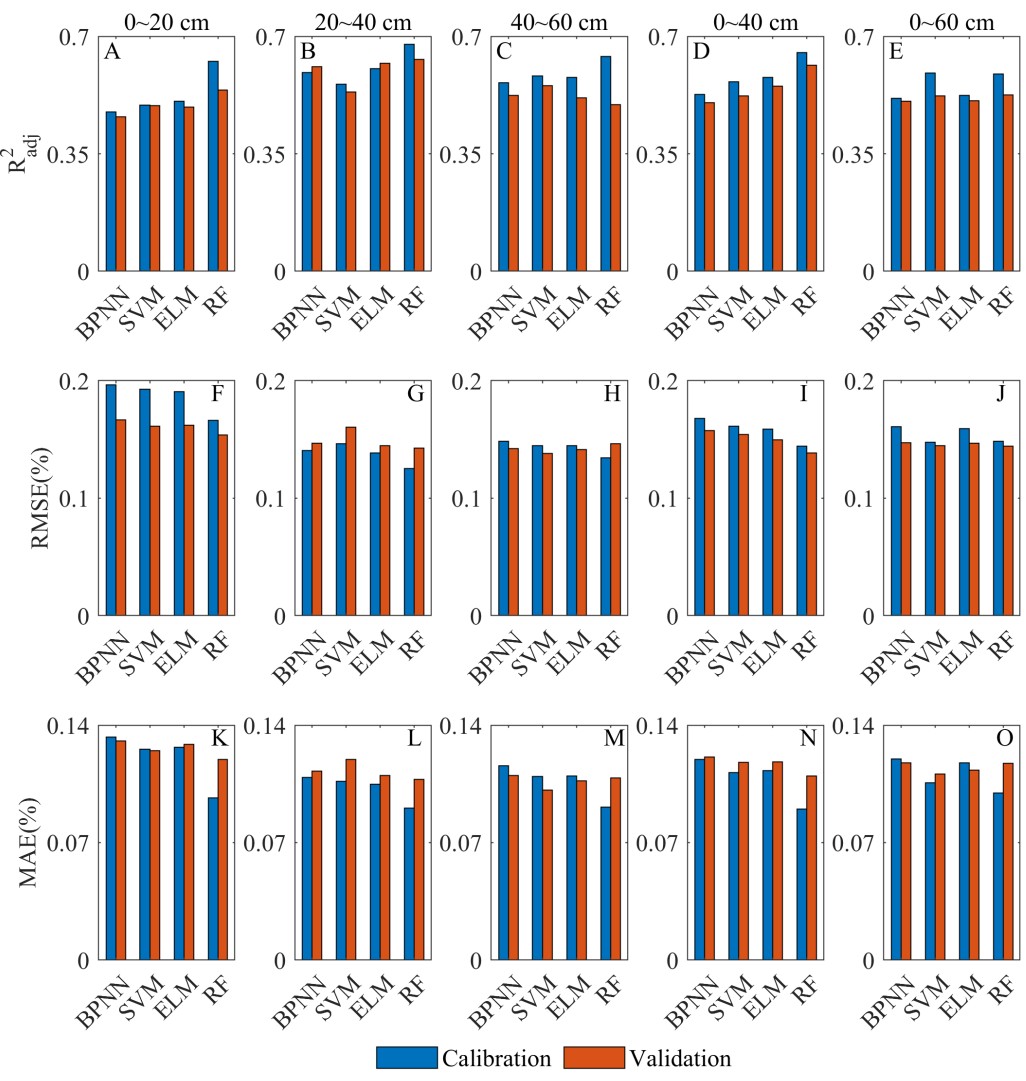

**Figure 8** **Overall comparative chart of SSC inversion depths.** (A–E) are $R^2_{adj}$ of the models at 0∼20 cm, 20∼40 cm, 40∼60 cm, 0∼40 cm and 0∼60 cm, respectively. (F–J) RMSE of the models at 0∼20 cm, 20∼40 cm, 40∼60 cm, 0∼40 cm and 0∼60 cm, respectively. (K–O) MAE of the models at 0∼20 cm, 20∼40 cm, 40∼60 cm, 0∼40 cm and 0∼60 cm, respectively.

indicated that all models were able to control the inversion errors at 20∼40 cm. Therefore, 20∼40 cm was the optimal depth for the Sentinel-2 data for SSC inversion. The $R^2_{adj}$ at 0∼20 cm and 0∼60 cm was relatively low overall, and the inversion error at 0∼20 cm was the highest among all depths (Fig. 8). The overall fitting at 40∼60 cm and 0∼40 cm was satisfactory, and the inversion error at 40∼60 cm was better controlled (Fig. 8). Therefore, Sentinel-2 had the lowest sensitivity at 0∼20 cm. At 0∼60 cm, it was slightly better than 0∼20 cm. The inversion performance was good at 40∼60 cm and 0∼40 cm, and that at 40∼60 cm was relatively better.

As is analyzed above, the sensitivity of Sentinel-2 to SSC at different depths in the vegetated area was as follows: 20∼40 cm > 40∼60 cm > 0∼40 cm >0∼60 cm >0∼20 cm.

*Evaluation of inversion models*

In this section, the SSC inversion ability of each model was evaluated by analyzing the combined performance at all depths of each model (Fig. 9). The overall $R^2_{adj}$ of the RF model was higher than that of the other models (Figs. 9A–9D). Although some overfitting occurred at 40∼60 cm, the model still had a good fitting and generalization performance. The RF model had the lowest RMSE and MAE at all depths (Figs. 9E–9L), and the inversion error was especially well controlled at 0∼20 cm where the inversion errors of other models were all high. Therefore, the RF model was the optimal model for SSC inversion. The BPNN model fitted relatively poorly at most depths (Figs. 9A–9D), and the MAE of BPNN model was high at all depths (Figs. 9I–9L). Therefore, the BPNN model had a relatively slightly poor SSC inversion ability. The SVM and ELM models had better fitting and generalization performance, and their inversion errors were relatively low (Fig. 9). The ELM model was obviously better for the inversion at 20∼40 cm than that of SVM model. Therefore, both of them have satisfactory SSC inversion performance though ELM model was slightly better.

When $R^2_{adj}$, RMSE and MAE were all taken into consideration, the SSC inversion capability of all models was as follows: RF model > ELM model > SVM model > BPNN model.

## SSC distribution of JIA

The optimal model (RF) was used to estimate the SSC distribution at 5 depths (0∼20 cm 20∼40 cm 40∼60 cm ∼40 cm and 0∼60 cm) of JIA (Fig. 10). The study area was interspersed with salinized soil in different degree. It was dominated by non-saline (SSC < 0.2%) and slightly saline soil (0.2% < SSC < 0.5%). The severely saline (0.5% < SSC < 1%) and saline soil (SSC > 1%) only account for a small portion and mainly distributed in the northwest of the area. The salinization in the south of the area was lower than that in the north, which may be related to the irrigation method of JIA (the water was drained from the south to the north, and salt was accumulated in the north). There was more slightly and severely saline soil at 0∼20 cm than at 20–40 cm and 40–60 cm, and more non-saline soil at 20∼40 cm than at other depths. Overall, the estimated SSC distribution of JIA in this study was consistent with the actual measured information (*Huang et al., 2018*).

## DISCUSSION

In this study, the full subset selection was used to select the optimal combination of IVs (included 12 bands and 32 spectral indices) at five depths, which mitigated the subjectivity of IV selection. In addition, $R^2_{adj}$ was used to evaluate the fitting so as to mitigate the difference among the coefficient of determination caused by the number of IVs. Therefore, the reliability of the comparison of inversion performance at different depths was improved. This study showed that Sentinel-2 was most sensitive to SSC at 20∼40 cm, followed by 40∼60 cm, and the sensitivity at other depths from high to low was 0∼40 cm, 0∼60 cm and 0∼20 cm. A similar result was obtained by *Zhang et al. (2019b)* when studying the sensitivity to SSC at different depths based on GF-1. It has been found that SSC around crop roots can affect the crop growth by producing osmotic stress (*Chen et al., 2003*). When studying the water absorption model of crop root system in salinization soil in HID, *Qiao*

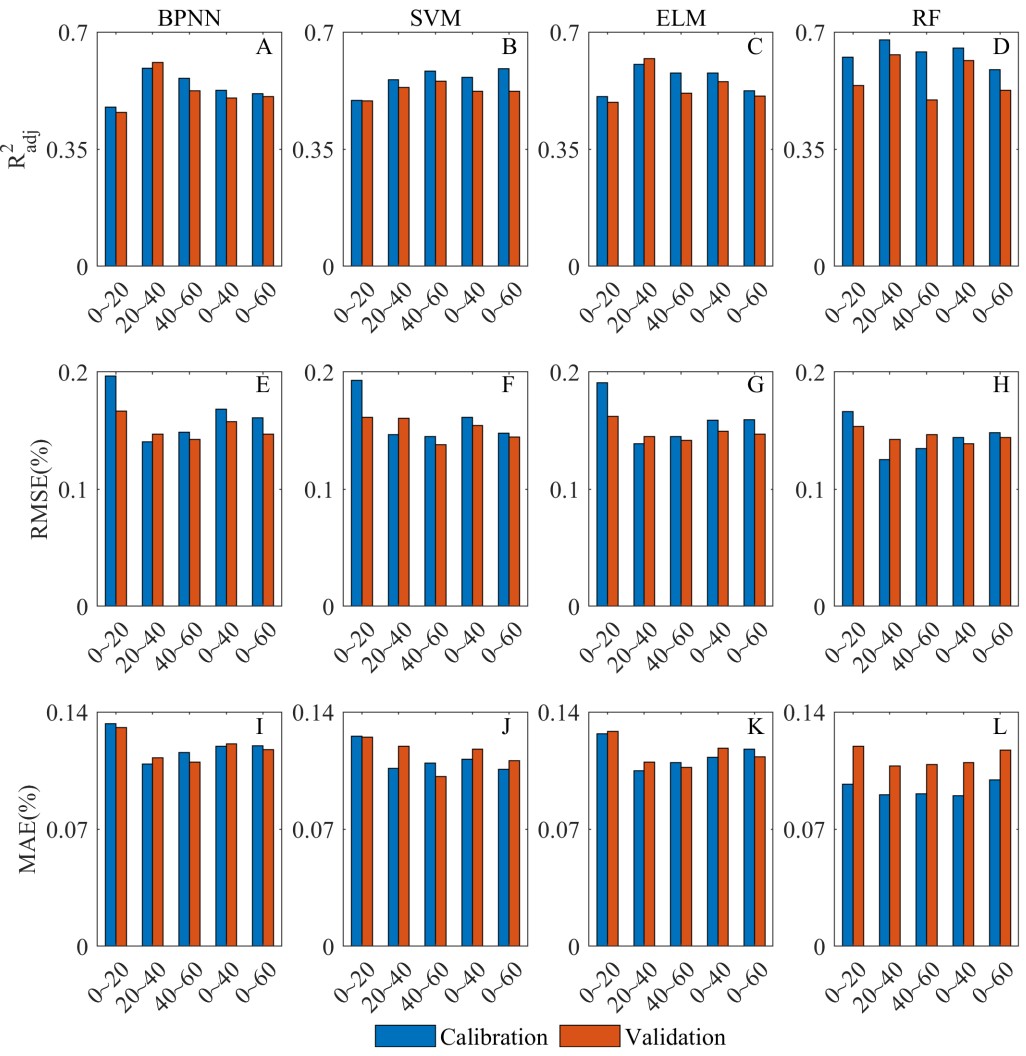

**Figure 9** **Overall comparative chart of SSC inversion models.** (A–D) are the $R^2_{adj}$ of BPNN, SVM, ELM and RF model at all depth, respectively. (E–H) RMSE of BPNN, SVM, ELM and RF model at all depth, respectively. (I–L) MAE of BPNN, SVM, ELM and RF model at all depth, respectively.

*(2005)* found that the main water absorption layer of sunflowers (accounting for 70% of the samples in our study) was at 0∼50 cm, and the peak of maximum water absorption was at 20∼40 cm. When the sunflower was in bloom (mid to late August), the surface soil moisture content could not meet the root demand, and the peak of maximum water absorption shifted to 35 cm. During this period, the water absorption rate at 20∼40 cm and 40∼60 cm was 2 and 1.5 times of that at 0∼20 cm, respectively, and the water increment mainly came from the deep soil layer. Therefore, the SSC at 20∼40 cm had the strongest effect on crop growth (crop growth can be reflected indirectly via remote sensing data), followed by 40∼60 cm, which was basically consistent with the results in our study.

Four machine learning algorithms (BPNN, SVM, ELM, and RF) were used for SSC inversion, and the RF model was found to perform well at all depths and was the optimal

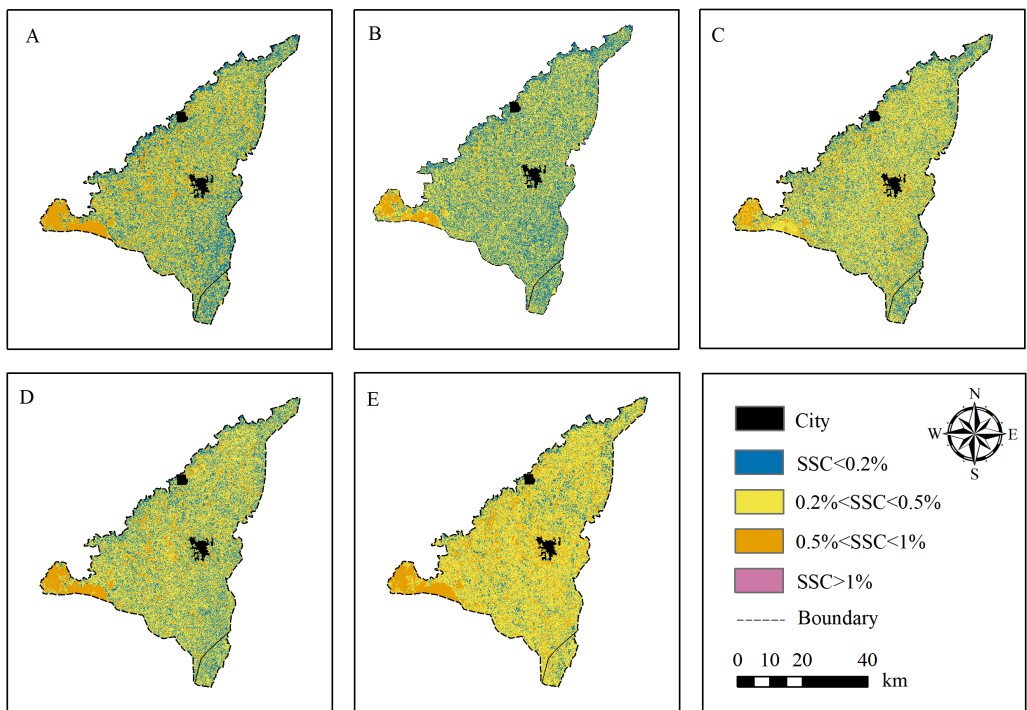

**Figure 10** **SSC distribution maps of JIA based on RF model.** (A–E) The SSC distribution at 0∼20 cm, 20∼40 cm, 40∼60 cm, 0∼40 cm and 0∼60 cm, respectively.

model for SSC inversion. This is due to the fact that the RF model is a collection of decision trees, which enables RF to have a good generalization performance and effectively limit the overfitting without reducing the prediction accuracy. The RF model has been found to have a good control over the noise (*Belgiu & Drăguţ, 2016*; *Taghizadeh-Mehrjardi et al., 2020*). Therefore, although the variability of SSC was high (CV was about 80%), the model could well cope with the outliers and avoid local optimal solutions. By comparing the SSC inversion accuracy used BPNN, SVM, multiple linear regression (MLR) and RF methods, *Zhang et al. (2019a)* found that the RF model performed optimally. Others (*Pahlavan-Rad et al., 2020*; *Chagas et al., 2016*; *Li et al., 2020*) have also obtained satisfying results when using RF model for prediction.

However, due to the different responding mechanisms to SSC at different crop growth period, and the effect of different crop planting structure on the inversion, the conclusions of this study to some extent only apply to the crop planting structure and growth period of this experiment. In addition, this study was limited to the analysis of the statistical relationship between SSC and spectral reflectance, and the responding mechanism between the two needs to be analyzed in more depth. Further studies can focus on evaluating the model performance at different crop growth period and structure, and analyzing the response mechanism of SSC, vegetation and spectrum.

# CONCLUSIONS

This study evaluated the sensitivity of Sentinel-2 to SSC at different depths in the vegetated area, and obtained the optimal estimating model. It can provide a certain basis for soil salinization monitoring in HID.

1. By analyzing the combined performance of all models (after full subset selection) at each depth, we found Sentinel-2 was most sensitive to SSC at 20~40 cm, the $R^2_{adj}$ of each model was around 0.6, which was significantly better than that at the other depths. It was the worst at 0~20 cm, but the $R^2_{adj}$ could also reach 0.45 in each model. The sensitivity of Sentinel-2 to SSC at different depths in the vegetated area was as follows: 20~40 cm > 40~60 cm > 0~40 cm > 0~60 cm > 0~20 cm.

2. According to the analysis of the combined performance at all depths of each model, we found that all four machine learning models have achieved good inversion results ($R^2_{adj} > 0.46$). The RF was the optimal model for SSC inversion. It had obvious advantages in both fitting and inversion accuracy, with $R^2_{adj}$ between 0.5~0.68 at all depths. The SSC inversion ability in the vegetated area of all models was as follows: RF model > ELM model > SVM model > BPNN model.

# ACKNOWLEDGEMENTS

The authors would like to thank the Key Laboratory of Agricultural Soil and Water Engineering in Arid and Semiarid Areas of Ministry of Education for providing the test equipment. The authors are especially grateful to the reviewers and editors for appraising our manuscript and for offering instructive comments.

## Funding

This research is supported by the National Natural Science Foundation of China (51409221, 51349001) and the National Key Research and Developement Program of China (2017YFC0403302). The funders had no role in study design, data collection and analysis, decision to publish, or preparation of the manuscript.

## Grant Disclosures

The following grant information was disclosed by the authors:
National Natural Science Foundation of China: 51409221, 51349001.
National Key Research and Developement Program of China: 2017YFC0403302.

## Competing Interests

The authors declare there are no competing interests.

## Author Contributions

- Yinwen Chen and Yuanlin Qiu conceived and designed the experiments, analyzed the data, prepared figures and/or tables, authored or reviewed drafts of the paper, and approved the final draft.

- Zhitao Zhang conceived and designed the experiments, authored or reviewed drafts of the paper, and approved the final draft.
- Junrui Zhang conceived and designed the experiments, performed the experiments, analyzed the data, authored or reviewed drafts of the paper, and approved the final draft.
- Ce Chen analyzed the data, authored or reviewed drafts of the paper, and approved the final draft.
- Jia Han conceived and designed the experiments, performed the experiments, authored or reviewed drafts of the paper, and approved the final draft.
- Dan Liu analyzed the data, prepared figures and/or tables, and approved the final draft.

## Field Study Permissions

The following information was supplied relating to field study approvals (i.e., approving body and any reference numbers):

The Hetao Irrigation District administration gave field permit approval to us (No. 2017YFC0403302).

## Data Availability

Measured soil salt content, band, spectral indices data are available as a Supplemental File.

## Supplemental Information

Supplemental information for this article can be found online at http://dx.doi.org/10.7717/peerj.10585#supplemental-information.

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
