# Peer review of "Estimating salt content of vegetated soil at different depths with Sentinel-2 data"

_PeerJ, doi:10.7717/peerj.10585_

## Round 0.1 · original submission · Major Revisions

This is an interesting manuscript that requires several improvements before it can be considered for publication. I kindly ask you to read carefully the recommendations given by reviewers before returning the revised version to PeerJ.

·

Basic reporting

Clear english style

Experimental design

original paper

Validity of the findings

the models were sound and statistically validated

Additional comments

In the present study, the authors have applied and compared four machines learning models namely, back propagation neural network (BPNN), support vector machine (SVM), extreme learning machine (ELM) and random forest (RF), for predicting soil salt content (SSC) measured at five different depth: (0~20 cm, 20~40 cm, 40~60 cm, 0~40 cm, and 0~60 cm) in the vegetated area. One of the novelties of the present paper is that, the authors reported that predicting SSC at vegetated area is rarely reported in the literature. For developing the models, the authors have used 12 bands and 32 spectral indices based on the Sentinel-2 (Sentinel-2A and 2B) data, as predictor (i.e., independent variables). In addition, the inversion capability of the models was investigated. In total, 117 soil samples were collected from Jiefangzha Irrigation Area (JIA), Inner Mongolia, China. Several well-known performances metrics were used for testing the capabilities of the models: the adjusted (R2), root mean square error (RMSE) and mean absolute error (MAE).

This is an interesting paper, well written and well-structured with a solid scientific research objective, constituting this way an excellent starting point of research paper. The paper needs only fewer amendments before to be accepted.

1. The introduction should be slightly amended. At the end of the line 102, provide a review of machines learning (ELM, SVM, RF and BPNN) applied in many area of water resources management, for example, for modeling water quality, i.e., water temperature, dissolved oxygen, and for many agricultural application (modeling of salinity, ET0…).
2. Provide the scatterplot of measured and calculated SSC using the best four machine learning models.
3. Improve the theoretical description of the machines learning models, and as possible provide a figure for each model.
4. Provide a flowchart of the proposed methodology.

Reviewer 2 ·

Basic reporting

no comment

Experimental design

no comment

Validity of the findings

no comment

Additional comments

Review of manuscript 51893, entitled “Estimating salt content of vegetated soil at different depths with Sentinel-2 data” by Y. Chen, Y. Qiu, Z. Zhang, J. Zhang, C. Chen, J. Han, and D. Liu.
This study presents four different machine learning methods for assessing soil salinity in Hetao, China, based on remote sensing data from the Sentinel-2 sensor. The topic certainly merits authors attention as soil salinity is one of the major soil degradation processes in the world. However, authors need to better explain how their study compares with research carried out on other places. They have missed some important references. They have also to improve the level of language used in the text. My suggestion is thus that the paper be sent for major revision.
General comments:
1. The language level needs to be improved. Besides proofreading by a native English speaker, authors need to review the vocabulary sometimes used as it is inadequate for a scientific paper. See L133 as an example where the term “nothing” was used instead of “bare soil”.
2. The introduction section needs to be improved by presenting the different methodologies used for salinity assessment via remote sensing (see comments below).
3. There is a general use of acronyms along the text without explaining their meaning. This needs to be corrected.
Specific comments:
L79-89: More important than referring the different satellites used, authors should focus on the methods and assumptions behind the studies cited here. They are quite different. Authors should also refer to the studies conducted by Lobell et al. (2010), Wu et al. (2014), and Scudiero et al. (2014), which constitute important references in this topic. Additionally, they should consider Ramos et al. (2020) as a first example where Sentinel-2 data was used to assess soil salinity.
L83: Authors should explain what they mean by inversion modelling somewhere in the text.
L88: Some of the advantages of Sentinel-2 are yet to be seen. Spatial resolution seems valid. About temporal resolution not so clear since only multi-year data has been considered so far in salinity assessment studies using Sentinel-2 data. Finally, the spectral offer is vast but only the bands in the visible and NIR range have proven to be related to salinity data. These are offered by most satellite platforms.
L94: Authors need to give the meaning of these acronyms. Also, for L107.
L97-99. This is not true. See Ramos et al. (2020) where Sentinel-2 data was used to assess rootzone salinity.
L127-128: Why is this relevant for a scientific paper?
L128-129: Bui and Henderson (2003) apparently worked in Queensland, Australia. How conditions and practices there relate to your region? This citation makes no sense.
L147: Please briefly explain how this formula was developed. Was that study conducted in Hetao?
L158: Each Sentinel-2 satellite may have a revisiting period of 10 days, but combined that period shortens to 5 days. That should be emphasised instead of the 10 days.
L161: Authors should limit themselves to the bands used in this study. For sure, bands 1, 9 and 10 were not used for salinity assessment.
L166-168: As far as I am aware, the Copernicus Hub already provides the 2019 Sentinel-2 data as a Level-2A product. This step could likely have been skipped.
L168-169: Please elaborate further on the resampling technique used.
L177: Please explain cleared what a “soil line” is. I’m not sure if I understood it.
L178: All acronyms should be given in full the first time they are used in the text.
L210-253: What does BPNN, SVM, ELM, and RF stand for?
L418-419: High SSC can increase the osmotic stress, not water stress. These are different processes. Please correct.
References used:
Lobell, D.B., Lesch, S.M., Corwin, D.L., Ulmer, M.G., Anderson, K.A., Potts, D.J., Doolittle, J.A., Matos, M.R., Baltes, M.J., 2010. Regional-scale assessment of soil salinity in the Red River Valley using multi-year MODIS EVI and NDVI. J. Environ. Qual. 39 (1), 35–41. https://doi.org/10.2134/jeq2009.0140
Ramos, T.B., Castanheira, N., Oliveira, A.R., Paz, A.M., Darouich, H., Simionesei, L., Farzamian, M., Gonçalves, M.C., 2020. Soil salinity assessment using vegetation indices derived from Sentinel-2 multispectral data. Application to Lezíria Gande, Portugal. Agric. Water Manag. 241, 106387, https://doi.org/10.1016/j.agwat.2020.106387
Scudiero, E., Skaggs, T.H., Corwin, D.L., 2014a. Regional scale soil salinity evaluation using landsat 7, western San Joaquin Valley, California USA. Geoderma Reg. 2–3, 82–90. https://doi.org/10.1016/j.geodrs.2014.10.004
Wu, W., Al-Shafie, W., Mhaimeed, A., Ziadat, F., Nangia, V., Payne, W., 2014. Soil salinity mapping by multiscale remote sensing in Mesopotamia. Iraq. IEEE J. Sel. Top. Appl. 7, 4442–4452. https://doi.org/10.1109/jstars.2014.2360411

---

## Round 0.2 · accepted · Accept

Considering that all issues raised by the reviewers were properly and carefully addressed by the authors with the improvements as required, it is our opinion that the manuscript can be accept in its present version.

Reviewer 2 ·

Basic reporting

no comment

Experimental design

no comment

Validity of the findings

no comment

Additional comments

I believe authors have addressed my previous comments in a satisfactory way and that the paper can be now accepted for publication.